# High−precision measurements of nitrous oxide and methane in air with cavity ring-down spectroscopy at 7.6μm

Jing Tang, Bincheng Li, Jing Wang

School of Optoelectronic Science and Engineering, University of Electronic Science and Technology of China, Chengdu 610054, China

*Correspondence to*: Bincheng Li (bcli@uestc.edu.cn)

**Abstract.** A high-sensitivity methane ($CH_4$) and nitrous oxide ($N_2O$) sensor based on mid-infrared continuous-wave (cw) cavity ring-down spectroscopy (CRDS) technique was developed for environmental and biomedical trace gas measurements. A tunable external-cavity mode-hop-free (EC-MHF) quantum cascade laser (QCL) operating at 7.4 to 7.8 µm was used as the light source. The effect of temperature fluctuation on the measurement sensitivity of the CRDS experimental setup was analyzed and corrected, and a sensitivity limit of absorption coefficient measurement of $7.2\times10^{-10}$ cm$^{-1}$ was achieved at 1330.50 cm$^{-1}$ with an average of 139 measurements, or 21-seconds averaging time, and further improved to $2.3\times10^{-10}$ cm$^{-1}$ with average of 3460 measurements, or 519-seconds averaging time. For the targeted $CH_4$ and $N_2O$ absorption lines located at 1298.60 cm$^{-1}$ and 1327.07 cm$^{-1}$, with temperature effect correction detection limits of 13 pptv and 11 pptv were experimentally achieved with 10.4-seconds and 10.2-seconds averaging time, and could be further improved to 5 pptv and 9 pptv with 482.5-seconds and 311-seconds averaging time, respectively. Four spectral bands (1298.4 cm$^{-1}$ to 1298.9 cm$^{-1}$, 1310.1cm$^{-1}$ to 1312.3cm$^{-1}$, 1326.5 cm$^{-1}$ to 1328 cm$^{-1}$, and 1331.5 cm$^{-1}$ to 1333 cm$^{-1}$) in the spectral range from 1295cm$^{-1}$ to 1335cm$^{-1}$ were selected for the separate and simultaneous measurements of $CH_4$ and $N_2O$ under normal atmospheric pressure, and all were in good agreements. The concentrations of $CH_4$ and $N_2O$ of atmospheric air collected at different locations and of exhaled breath were measured and analyzed. Continuous measurements of $CH_4$ and $N_2O$ concentrations of in-door laboratory air over 45 hours was also performed. It was found that anaerobic bacteria in water and soil of wetland might significantly increase the $CH_4$ concentration in air. The measured $N_2O$ concentration in the central city area was somewhat lower than the reported normal level in open air. Our results demonstrated the temporal and spatial variations of $CH_4$ and $N_2O$ in air.

## 1 Introduction

Methane ($CH_4$) and nitrous oxide ($N_2O$) are two of the most important atmospheric greenhouse gases, of which the concentrations are rising up continuously since pre-industrial time (Hartmann *et al*., 2013). Moreover, the global warming potential (GWP) of $CH_4$ is about 25 times greater than that of carbon dioxide ($CO_2$) (Boucher *et al*., 2009), while the GWP of $N_2O$ is 300 times (Rapson and Dacres, 2014) greater than that of $CO_2$. Apart from natural processes, the spatial distribution of both, to a great extent, depends on human-being activities, such as agricultural practices (Mosier *et al*., 1998), organic waste,

industrial activities, and so on. Even small changes of concentrations of $CH_4$ and $N_2O$ in atmosphere are of great influence to natural environment. Therefore, the highly sensitive and precise measurements of $CH_4$ and $N_2O$ concentrations in atmospheric air are essential to environmental monitoring and greenhouse gas controlling. For sensitive $CH_4$ and $N_2O$ detection in air a spectral range around 7.6μm is one of the most suitable spectral bands, as (1) in the wavenumber range from $1290cm^{-1}$ to

$1350cm^{-1}$ $CH_4$ and $N_2O$ have the second strongest fundamental vibration bands, and (2) in this spectral range there are minimum interference absorption lines from other gases (such as carbon dioxide ($CO_2$), carbon monoxide (CO), ammonia ($NH_3$), nitrogen monoxide (NO), etc.) in air except water vapor, which can be easily eliminated by drying the gas under test.

Cavity enhanced absorption techniques, such as cavity ring-down spectroscopy (CRDS) (Banik *et al*., 2017), integrated cavity output spectroscopy (ICOS) (O'Keefe, 1998), noise immune cavity enhanced optical heterodyne molecular spectroscopy

(NICE-OHMS) (Foltynowicz *et al*., 2008) and so on, have been wildly applied in sub-ppm- and even sub-ppb-level trace gas detections. The CRDS technique was first introduced by O'Keefe in 1988 (O'Keefe *et al.*, 1988), while now many commercial instruments based on CRDS have been developed for various applications, mostly for trace gas detections and real-time monitoring. Generally, due to the use of high-finesse cavity, the equivalent absorption length of CRDS instruments is thousands to tens of thousands of times longer than that of direct absorption spectroscopy using the same-length sample cell (Romanini,

1997), therefore the measurement sensitivity of CRDS-based instruments is much improved (more than three orders of magnitude) compared to that of direct absorption spectroscopy measurements. Furthermore, compared with the traditional chemical detection methods, such as gas chromatography (GS) (Loftfield *et al*., 1997) and mass spectrometry (MS) (De Gouw *et al*., 2003), CRDS is allowed to perform real-time measurements under the premise of high-sensitivity without time-consuming sample preparations. As both high sensitivity and real-time detection are of great significance to environmental

monitoring, CRDS is a suitable method for atmospheric trace gas monitoring. Moreover, CRDS also has the potential for exhaled breath test (Mashir and Dweik, 2009), since the exhaled breath contains many biomarker trace gases (for example $CH_4$ (De Lacy Costello *et al*., 2013), NO (Brubaker, 2016), $N_2O$ (Bleakley and Tiedje, 1982), $NH_3$ (Kearney *et al*., 2002), etc.) that reflect some physiological processes and/or diseases in human body. However, mid-infrared (Mid-IR) CRDSs for trace gas detections were rarely reported in early days because of the unavailability of Mid-IR laser sources. Mid-IR light sources

based on nonlinear optical techniques, such as quasi-phase matching difference frequency generation (QPM-DFG) (Petrov *et al*., 1996), had too low output power, e.g. 16 μW (Whittaker *et al*., 2012), to have practical applications. In recent years, with the rapid development of advanced tunable high-power mid-infrared sources, especially external-cavity quantum cascade lasers (EC-QCL) (Botez *et al*., 2018), the LODs of CRDS for trace gas detections have been greatly improved. For example, Maity *et al.* (2017) achieved LOD of 52 pptv for $CH_4$ at 7.5 μm, Banik *et al.* (2017) achieved LOD of 5 ppbv for $N_2O$ at 5.2

μm, Long *et al*. (2016) achieved LOD of 2 pptv for $N_2O$ at 4.5 μm,  Maithani *et al*. (2018) achieved LOD of 740 pptv for $NH_3$ at 6.3 μm and Zhou *et al*. (2018) achieved LOD of 410 pptv for $NH_3$ at 5.3 μm.

In this paper, we developed a trace gas sensor based on Mid-IR cw-CRDS technique with a tunable EC-MHF QCL operating at the spectral range from 1290 to 1350 cm$^{-1}$ and applied the setup to detect trace $CH_4$ and $N_2O$ in normal laboratory air and out-door atmospheric air as well as in exhaled breath. Experimentally it was observed that the measurement results were subject to a temperature fluctuation of about 0.4°C caused by air conditioning for the laboratory room where the measurements were performed. This effect of temperature fluctuation on CRDS measurements was analysed in detail and corrected via data processing, which resulted in an improvement in the measurement sensitivity of CRDS. With the correction of temperature effect, a measurement sensitivity as low as $7.2 \times 10^{-10}$ cm$^{-1}$ absorption coefficient was experimentally achieved. To achieve high measurement sensitivity as well as high reliability for separate and simultaneous detections of trace $CH_4$ and $N_2O$ in atmospheric air under normal atmospheric pressure, four wavenumber bands within the spectral range of the QCL were selected for the reliable concentration determinations of $CH_4$ and $N_2O$, with one band for separate $N_2O$ detection, two bands for separate $CH_4$ detection, and one band for simultaneous $CH_4$ and $N_2O$ detections. The $CH_4$ and $N_2O$ concentrations determined from the four bands were in good agreement, indicating the reliability of the measurement results. Finally, the developed CRDS experimental setup was used to measure the concentrations of $CH_4$ and $N_2O$ collected at different locations as well as one collected exhaled breath, and to monitor simultaneously $CH_4$ and $N_2O$ concentrations of in-door laboratory air continuously for over 45 hours, demonstrating the applicability of CRDS for sensitive environmental monitoring and exhaled breath analysis.

## 2 Experimental setup

The CRDS experimental setup is schematically depicted in Fig.1. A tunable Mid-IR external-cavity CW-MHF QCL (41074-MHF, Daylight Solutions) is used as the optical source, which outputs continuously a collimated laser beam with a narrow linewidth (<30MHz or 0.001 cm$^{-1}$) and a relatively high power ($\sim$160mW) in the spectral range from 1290cm$^{-1}$ to 1350cm$^{-1}$. To block the reflection of the laser beam by the ring-down cavity optics from re-entering the QCL resonator and de-stabilizing the output spectrum and power, an optical isolator with central wavelength of 7.2 μm and isolation ratio of $>$ 30 dB (FIO-5-7.2, Innpho) is placed in front of the laser output port. Subsequently, the QCL beam propagates through an acousto-optic modulator (AOM, acting as a fast optical switch) (I-M041, Gooch & Housego) controlled by a home-made high-speed (with response time $<$50ns) threshold trigger, and the first-order beam outputted from the AOM is coupled into the ring-down cavity (the sample cell) consisting of a 50-cm-long stainless steel tube (CRD Optics). A pair of high-reflectivity (reflectivity $>$99.98%; CRD Optics) plane-concave mirrors with diameter of 1 inch and radius of curvature of -1 meter are installed at both ends of the sample cell via two three-dimensionally adjustable optical mounts which are mounted to the sample cell by screws. A He-Ne laser at 632.8nm is employed to help aligning the high-reflectivity cavity mirrors. The QCL beam that transmitted through the sample cell is focused by a focusing lens, placed closely behind the rear cavity mirror, into a highly sensitive (detectivity of $2.5 \times 10^9$ cm$\sqrt{Hz}$/W at 8μm), TE-cooled, high-speed HgCdTe infrared photovoltaic detector (PVMI-4TE-8, Vigo, Poland). Then the detected CRD signal is recorded by a data acquisition (DAQ) card (M2i.3010, Spectrum Instrumentation, Germany)

and processed by a MATLAB program in real time. The QCL is tuned by the laser controller (via synchronously adjusting the tuning grating and the length of the laser cavity) with a step of 0.01 cm$^{-1}$ (with accuracy < 0.003 cm$^{-1}$). As the free-spectral-range (FSR, 300MHz or 0.01 cm$^{-1}$) of the ring-down cavity (RDC) is much larger than the laser linewidth (< 0.001 cm$^{-1}$), at each step the RDC length is modulated via three piezoelectric transducers (PZT, Model PE-4, Thorlabs) attached to the optical

mount installing the rear high-reflectivity cavity mirror. The PZTs are synchronously driven by a triangular wave function generated by a three-channel open-loop PZT driver (MDT694B, Thorlabs) to modulate periodically the RDC length over one half of the wavelength, about 4 μm, for the coupling of QCL laser power into the RDC via resonance of the laser spectral line with one RDC mode. Within one cavity length modulation period, laser power with a TEM$_{00}$ mode (RDC mode) builds up inside the RDC, correspondingly the beam power transmitted through the RDC and detected by the infrared detector also

increases rapidly. At the time the detected signal amplitude exceeds a pre-set voltage threshold (20mV-2000mV), the threshold trigger sends out a triggering signal to shut down the AOM and a ring-down signal sequence is recorded by the DAQ and processed by a personal computer (PC). A vacuum pump (nominal ultimate pressure < 8 mbar, MPC 301Z, Welch) and a pressure gauge (nominal pressure accuracy ±0.5 mbar, LEX1, Keller) are connected to the sample cell to control the pressure of gas mixture under test and to replace gas mixture inside the sample cell when necessary. During the laser spectral tuning,

at each step the frequency is determined by the RDC mode in resonance with the laser line, the maximum frequency error should be <0.01cm$^{-1}$ (determined by the FSR of RDC and the scan step), as the frequency at each RDC mode is not accurately controlled.

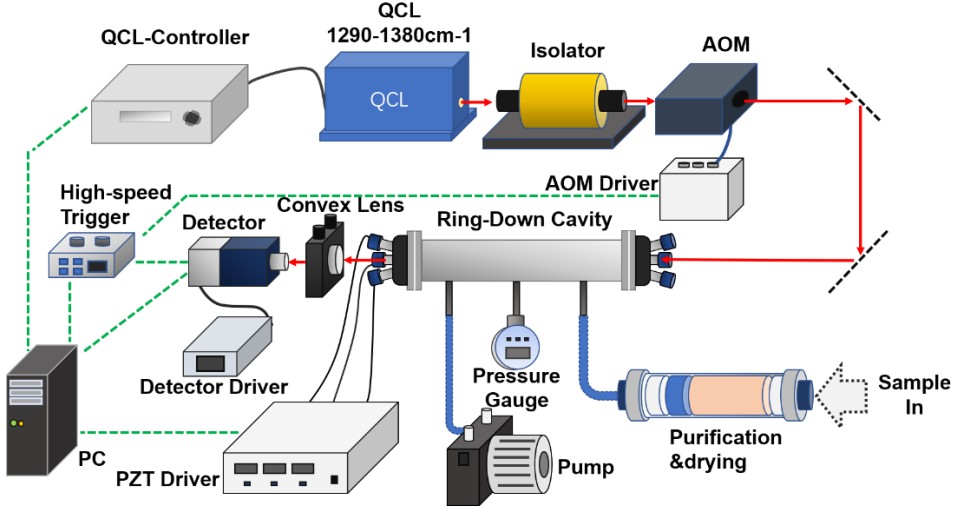

**Figure 1 Schematic diagram of CRDS experimental set-up**

A fitting program based on Levenberg-Marquardt algorithm is applied to fit the recorded ring-down signal to an exponential decay function to determine the ring-down time $\tau$. By tuning the QCL wavelength, the dependence of ring-down time on wavelength over the required spectral range is obtained. The wavelength-dependent absorption coefficient $\alpha$ of the gas sample within the sample cell is determined from the measured ring-downtime $\tau$ using equation $\alpha(\lambda) = \frac{1}{c}\left(\frac{1}{\tau} - \frac{1}{\tau_0}\right)$, where $c$ is the

speed of light, $\lambda$ is the laser wavelength, and $\tau_0$ is the ring-down time of an "empty" cavity (without absorbing sample inside the sample cell).

Since in normal atmospheric air the concentration of water vapor is in the range from 100 ppm up to 4%, and water vapor has strong absorption lines that makes no CRDS signals can be experimentally observed in the selected spectral range from

$1290cm^{-1}$ to $1350cm^{-1}$, before measurements the water vapor in the gas mixture under test has to be mostly removed to a very low level (<10ppm) which has negligible influence on the $CH_4$ and $N_2O$ measurements. In our experiment, a 3A molecular sieve (HuShi Ltd., China), which only allows molecules whose dynamic diameter is less than 0.3 nm (Ruthven, 1984), such as water vapor and ammonia, to be adsorbed on it, is employed as the desiccants to eliminate the water vapor in the gas mixture. A filter tube, which is served as the gas inlet of the sample cell, filled up with such desiccants and quartz cotton is connected

to the sample cell for purification and drying of the gas sample. In addition, a cage with the same desiccants is put inside the sample cell to absorb the water vapor leaked in, therefore to keep the sample cell nearly water vapor free. With these means for water vapor removal, the residual water vapor in the sample cell is below 1 ppmv, and can keep below 1 ppm for several months after one desiccant filling. With this drying method this CRDS experimental setup is capable of analysing both canned dry gas mixture and untreated atmospheric air with a moderate water vapor concentration. Experimental results demonstrate

the effectiveness of this drying process as no absorption lines of water vapor are observed in the measured spectra. On the other hand, a spectral line of $1312.5 \ cm^{-1}$ of the water vapor, as presented in section B (not shown), can be used to monitor or even determine simultaneously the water vapor concentration (below 100 ppmv) if necessary.

The gas mixtures used in the experiment are ambient air collected at different locations within the university campus in the central city area of Chengdu, China at the same period of time (3:00-5:00 p.m. June 13, 2018), three hours after a light rain.

One sample is the air from the laboratory room (A), one is from an out-door parking lot outside the laboratory building (B), and one is from a wetland in the campus (C). The exhaled breath (D) of one healthy male person is also collected in the laboratory room (same as A) for measurements. In-door laboratory air is also continuously measured over 45 hours (from November 6 to 8, 2018). In our experiment, the exhaled breath air is collected with a 3L sampling bag, which can be fully filled with only one deep breath of a participant. The filled sampling bag is then connected to the sample cell via a valve. The

sample cell is first vacuumed by the vacuum pump and then filled with the exhaled air by opening the valve. This procedure is repeated two times for a complete replacement of gas in the sample cell by the exhaled air. As the volume of the sample cell is around 0.5L, the exhaled air of the 3L sampling bag is sufficient for exhaled air measurement. Similar procedure for out-door open air collection is followed.

## 3 Results and discussions

### 3.1 Limit of detection without/with temperature fluctuation correction
For sensitive trace gas detections, the sensitivity limit of the CRDS experimental setup is first tested with an "empty" cavity.

In our case, the "empty" cavity is filled with normal laboratory air with a reduced pressure of 6.4 mbar (the lowest pressure reached by the vacuum pump) and measured at an absorption-free wavenumber (1330.50 cm$^{-1}$). Figure 2(a) presents the recorded ring-down time of the "empty" cavity over 4400 seconds and corresponding fast Fourier transform (FFT) spectrum of the ring-down time sequence. To improve the measurement sensitivity, in general the CRDS signal is averaged to enhance

the signal-to-noise ratio (SNR) of the measurements and an optimal averaging number is determined by Allan variance. Figure 3 shows the calculated Allan variance versus averaging number for the recorded "empty" ring-down time. The optimal averaging number is determined to be 151, corresponding to 22.5-seconds averaging time. With the optimal averaging number, the average empty ring-down time ($\tau_0$) is 13.1 μs with a standard deviation (1σ) of $4.2\times10^{-3}$ μs, which is translated to a minimum absorption coefficient ($\alpha_{min}$) of $8.1\times10^{-10}$ cm$^{-1}$. From Fig. 2(a) periodical fluctuations of the ring-down time are observed, as

clearly indicated in the low-frequency end of corresponding FFT spectrum. To investigate the sources for these low-frequency periodical fluctuations, the temperature in the laboratory room is recorded simultaneously and the results are presented in Fig. 2(b). The temperature also shows periodical fluctuations with frequencies of the periodical fluctuations of the ring-down time, as demonstrated by the FFT spectrum of the temperature. The results presented in Fig. 2(a) and (b) clearly indicate that there is a positive correlation between the periodical fluctuations of the ring-down time and temperature in the low-frequency end.

That is, the low-frequency periodical fluctuation of the measured ring-down time is partially caused by the temperature fluctuation in the laboratory room.

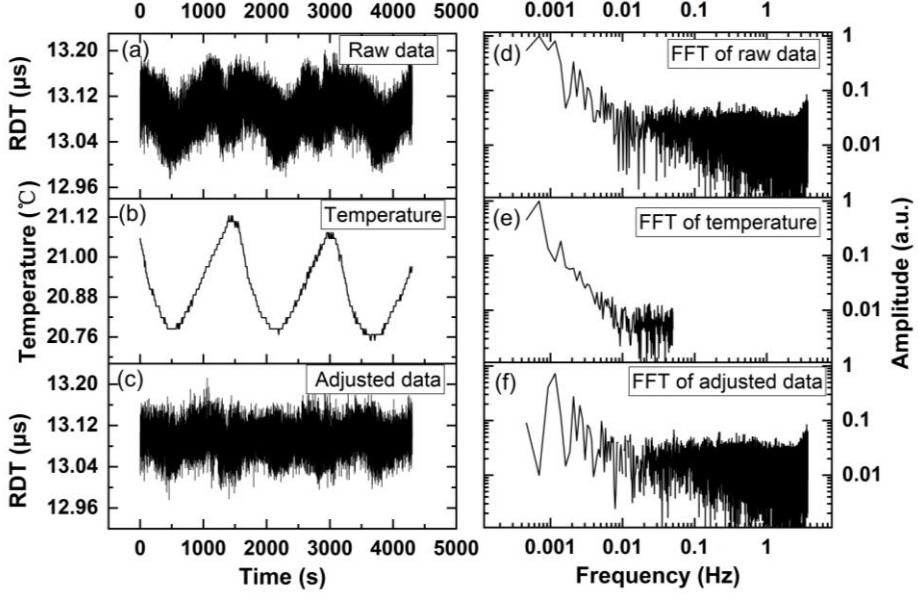

**Figure 2 (a) The "empty" ring-down time sequence recorded over 1 hour and (d) corresponding FFT spectrum. (b) The synchronously recorded temperature in the laboratory room and (e) corresponding FFT spectrum. (c) The "empty" ring-down time**

**sequence after the temperature effect is eliminated with the subtracting method and (f) corresponding FFT spectrum and. RDT represents "ring-down time".**

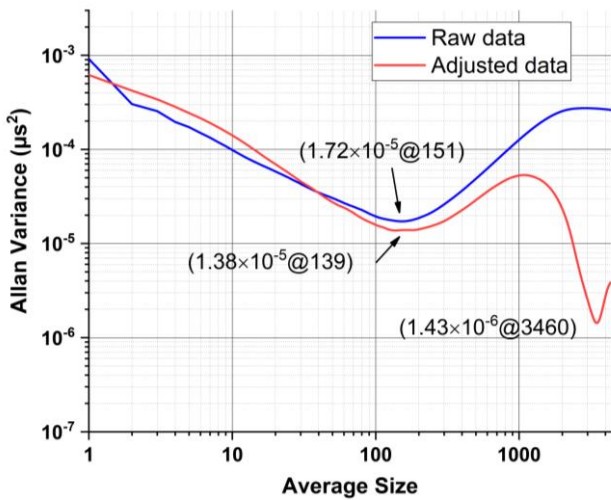

**Figure 3 Allan variance plot of raw and adjusted data (without/with temperature effect correction).**

A detailed investigation reveals that the correlation of ring-down time to temperature fluctuation is mainly caused by the different temperature dependence of the response of the three PZTs as well as the sensitivity of the ring-down time to the misalignment of the cavity mirrors. To test the sensitivity of the ring-down time to the alignment of the cavity mirrors, we first align the cavity mirrors to optimal positions, then apply an offset voltage to each PZT (while no offset voltage is applied to the other two PZTs) and observe how the measured ring-down time is influenced by the applied offset voltage. The results are presented in Fig. 4. An approximately linear relationship between the ring-down time and the applied offset voltage exists for each PZT, and the slopes of such linear dependences for different PZTs are different. This phenomenon is attributed to the PZT's difference in the creep and thermal-drift characteristics. As normally known, PZT is a nonlinear component and no two PZTs' characteristics are identical (Jaffe et al., 1971). Moreover, in our experiment the PZTs are controlled in an open-loop mode. Due to the different temperature sensitivity of the response of each PZT, temperature fluctuation causes misalignment of the cavity mirrors, which further results in a fluctuation in measured ring-down time, as presented in Fig. 2(a). Other thermal effects, such as the cavity length fluctuation, reflectivity fluctuation caused by temperature fluctuation, are negligible as compared to cavity alignment fluctuation. It is worth mentioning that experimental observation shows when the same offset voltage is synchronously applied to all three PZTs, the measured ring-down time is approximately independent on the voltage. This is the case when CRDS measurements are performed.

To eliminate the effect of temperature fluctuation on trace gas detection with CRDS, those frequency components in the FFT spectrum of ring-down time also presented in the FFT spectrum of the temperature fluctuation are subtracted mathematically and the ring-down time sequence is adjusted accordingly, as presented in Fig. 2(c). The subtraction is performed with both FFT spectra normalized to the frequency component with the maximum amplitude, which appears at the main frequency of the temperature fluctuation. This subtraction method is reasonable as at the main frequency of the temperature fluctuation, the

contributions of other factors to the fluctuation of the ring-down time are negligible as compared to that of the temperature fluctuation. After the effect of temperature fluctuation on the ring-down time measurement is eliminated, the absorption coefficient sensitivity limit $\alpha_{min}$ is first improved to $7.2\times10^{-10}$ cm$^{-1}$, with the optimal averaging number changes to 139, corresponding to 21 seconds averaging time, as presented in Fig. 3. Figure 3 also shows that there is a second minimum Allan variance if the averaging time is further increased, indicating $\alpha_{min}$ can be further improved to $2.3\times10^{-10}$ cm$^{-1}$ with optimal averaging number of 3460 and corresponding averaging time of 519 seconds. The results demonstrate that with temperature effect correction the measurement sensitivity could be greatly improved with a compromise of increasing measurement time (519 seconds versus 22.5 seconds).

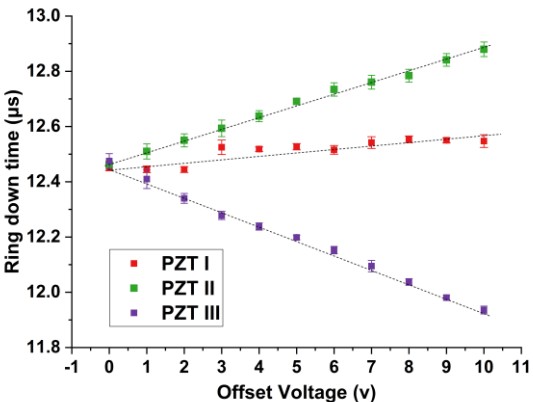

**Figure 4 Linear relationship between the offset voltage on each PZTs and the measured ring-down time.**

For real trace gas detections, the sensitivity limit achieved above with the "empty" cavity may not be fulfilled due to the presence of absorbing sample in the cavity and other effects such as laser wavelength fluctuation, limited wavelength tuning step for spectral measurement, etc. To find out the limits of detection (LOD) of $CH_4$ and $N_2O$ with the CRDS experimental setup, the sample cell is filled with ambient air at 1 atmospheric pressure and the ring-down data is recorded continuously at peaks of the absorption lines of $CH_4$ (1298.60 cm$^{-1}$) and $N_2O$ (1327.07 cm$^{-1}$), respectively, and the corresponding Allan variances are calculated. The achieved minimum $\sigma_{Allan}$ values are $3.1\times10^{-9}$ cm$^{-1}$ at 1327.07 cm$^{-1}$ for $CH_4$, and $2.8\times10^{-9}$ cm$^{-1}$ at 1298.60 cm$^{-1}$ for $N_2O$, which correspond to an LOD of 22 pptv for $CH_4$ and 16 pptv for $N_2O$, respectively. These LODs are obtained with approximate 15-seconds averaging time. The achieved LODs are lower than that achieved by other groups employing CRDS for $CH_4$ and $N_2O$ detections in recent years, as described in Sec. 1.

The LODs for $CH_4$ and $N_2O$ detections can be improved by eliminating the effect of temperature fluctuation via the process presented above. Again two Allan variance minima are present in the dependence of Allan variance on averaging time after the temperature effect is corrected, the corresponding LODs for $CH_4$ and $N_2O$ detections are 13 pptv and 11 pptv with 10.4-seconds and 10.2-seconds averaging time for the first minimum, and 5 pptv and 9 pptv with 482.5-seconds and 311-seconds

averaging time for the second minimum, respectively. Such low LODs allow sensitive detections of $CH_4$ and $N_2O$ with sub-ppbv-level concentrations.

### 3.2 Detection of $CH_4$ and $N_2O$ in ambient and exhaled breath air

For simultaneous detections of $CH_4$ and $N_2O$ in real applications, the optimal absorption lines or spectral ranges have to be carefully selected. For the spectral range from 1290$cm^{-1}$ to 1350$cm^{-1}$, $N_2O$ and $CH_4$ both have strong absorption lines. Figure 5 shows the spectral lines of $N_2O$ and $CH_4$ in the spectral range from 1295$cm^{-1}$ to 1335$cm^{-1}$ at 1 and 0.01 atmospheric pressures, respectively (The spectral data are from HITRAN 2016). When the pressure in the sample cell is reduced, individual absorption lines are well separated and can be fitted independently. At 1 atmospheric pressure, on the other hand, absorption lines are mixed and partially overlapped, especially when $CH_4$ and $N_2O$ are both presented. In this case care has to be taken to select appropriate spectral band(s) for separate or simultaneous detections of $CH_4$ and $N_2O$. In our experiment, four spectral sections in the spectral range from 1295$cm^{-1}$ to 1335$cm^{-1}$ are tested for the detections of $CH_4$ and $N_2O$. The selected spectral sections are listed in Table I. That is, Section A contains one $N_2O$ absorption line that is slightly weaker than the strongest $N_2O$ absorption line (1297.8315 $cm^{-1}$, $1.689\times10^{-19}$ $cm^{-1}$/(molec·$cm^{-2}$)) but is well separated from the adjacent absorption lines of $CH_4$ and $N_2O$. Section B contains three $N_2O$ lines and one $CH_4$ line. These four absorption lines are sufficiently strong and well separated. Sections C and D contain combinations of three and four overlapped absorption lines of $CH_4$ which are well separated from the absorption lines of $N_2O$ in the measurable spectral range. Section A is used for independent $N_2O$ detection, while Sections C and D are used for independent $CH_4$ detection, and Section B is for simultaneous $CH_4$ and $N_2O$ detections.

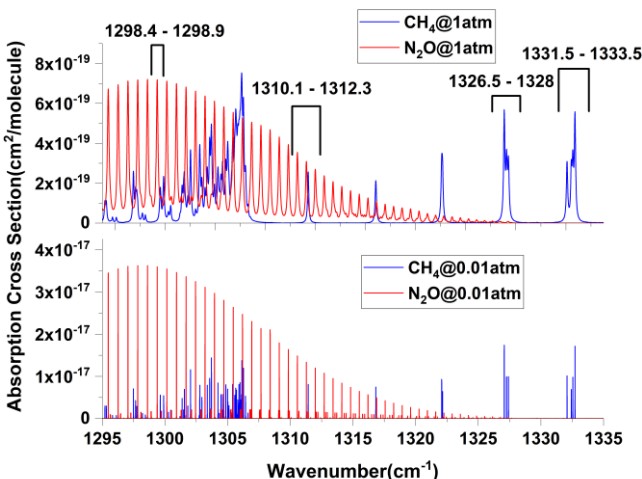

**Figure 5 HITRAN spectra of $N_2O$ and $CH_4$ in the spectral range from 1295 $cm^{-1}$ to 1335 $cm^{-1}$, at (a) 1 and (b) 0.01 atmospheric pressure, respectively.**

Table I. Selected spectral sections for simultaneous measurements of $CH_4$ and $N_2O$ in air

| Spectral Section | Spectral Range (cm$^{-1}$) | $CH_4$ absorption line /intensity (cm$^{-1}$ / cm$^{-1}$/(molec·cm$^{-2}$)) | $N_2O$ absorption line /intensity (cm$^{-1}$ / cm$^{-1}$/(molec·cm$^{-2}$)) |
|---|---|---|---|
| A | 1298.4 - 1298.9 | - | 1298.6031 / 1.681×10$^{-19}$ |
| B | 1310.1 - 1312.3 | 1311.431 / 4.447×10$^{-20}$<br>1311.2841 / 6.008×10$^{-20}$<br>1311.9973 / 5.403×10$^{-20}$ | 1310.5673 / 6.693×10$^{-20}$ |
| C | 1326.5 - 1328 | 1327.074 / 9.694×10$^{-20}$<br>1327.257 / 5.816×10$^{-20}$<br>1327.410 / 5.82×10$^{-20}$ | - |
| D | 1331.5 - 1333 | 1332.085 / 5.766×10$^{-20}$<br>1332.425 / 3.843×10$^{-20}$<br>1332.547 / 5.768×10$^{-20}$<br>1332.721 / 9.624×10$^{-20}$ | - |

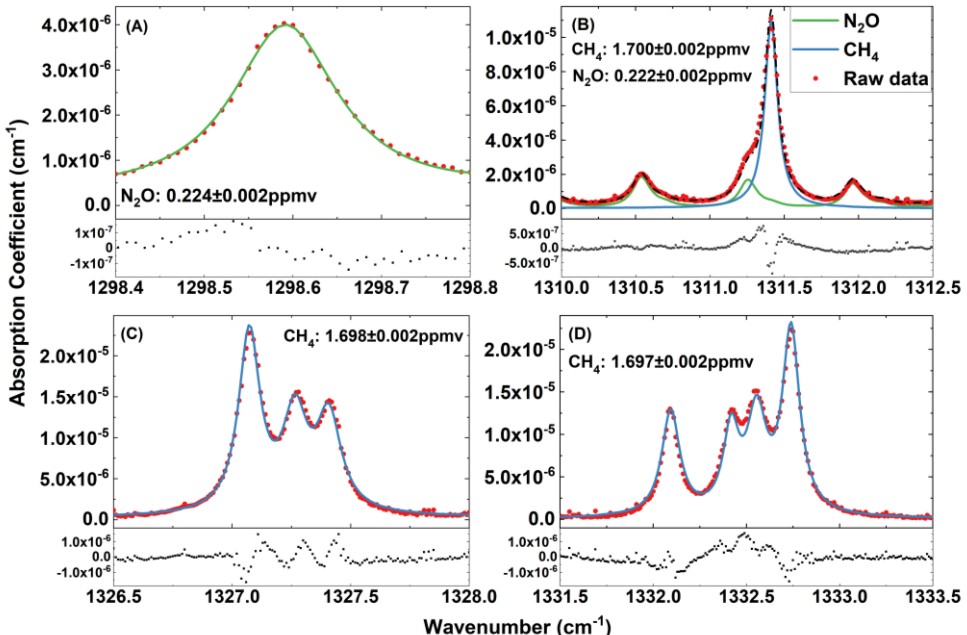

**Figure 6 Measured spectra (circles), corresponding best fits (solid lines), and fit residuals (lower figures) for four selected spectral bands (A: 1298.3-1299.1 cm$^{-1}$, B: 1310.1-1312.3 cm$^{-1}$, C: 1326.5-1328 cm$^{-1}$, and D: 1331.5-1333.5 cm$^{-1}$).**

Figure 6 shows the measured spectral lines, corresponding best fits, and fit residuals for the ambient air collected in the laboratory room. The measured data are the average of 128 measurements, took approximate 5 seconds for each wavenumber point. When performing the spectral fitting, the spectral profile is assumed to be Voigt, the laser frequency is linearly shifted to match the spectral lines of the target gas. From Section A, the $N_2O$ concentration is determined to be 0.224±0.002ppmv.

From Sections C and D, the $CH_4$ concentration is determined to be 1.698±0.002ppmv and 1.697±0.002ppmv, respectively, while from Section B, the $CH_4$ and $N_2O$ concentrations are determined to be 1.700±0.002ppmv and 0.222±0.002ppmv, respectively. The 2 ppbv concentration uncertainties represent the standard deviation of 6 repeat measurements. The small differences among the concentration values determined from different sections are mainly due to the misalignment caused by AOM induced small change in the deflection angle of the diffracting laser beam when tuning the laser wavenumber. As the RDC is aligned at 1310 cm$^{-1}$, in principle the concentrations obtained from section B are mostly close to the true values. Overall the $CH_4$ and $N_2O$ concentrations determined from different sections are well consistent, indicating the reliability of the measurement results. The good agreements between the $CH_4$ and $N_2O$ concentrations determined separately (from sections A, C, and D) and simultaneously (from section B) demonstrate that $CH_4$ and $N_2O$ concentrations can be simultaneously determined by employing a narrow band containing absorption lines of both gases for spectral measurements, therefore to shorten the measurement time.

The concentration uncertainties can also be estimated from the fit residuals presented in Fig. 6. The estimated uncertainties for $N_2O$ concentration are 4 ppbv from section A and 13 ppbv from section B, and for $CH_4$ concentration are 19 ppbv from section B, 19 ppbv from section C, and 18 ppbv from section D, respectively. These values are higher than 2 ppbv determined from repeat measurements due to the large fit residuals appeared around the absorption peaks, which are caused by uncertainties in wavelength, HITRAN spectral line intensity, line mixing (Gordon et al., 2017), pressure, and temperature, etc. Our calculations indicate wavelength uncertainty and HITRAN spectral line intensity error are the major sources for the large residuals around the absorption peaks. From the residuals departing from the peaks the estimated uncertainties for $N_2O$ concentration become 2 ppbv from section A and 4 ppbv from section B, and for $CH_4$ concentration are 3 ppbv from section B, 4 ppbv from section C, and 4 ppbv from section D, respectively. These uncertainty values become comparable to the 2ppbv determined from repeat measurements. As in principle CRDS measures the absolute absorption, the concentration uncertainties obtained from the spectral fit residuals represent the absolute accuracy for the concentration determination, the uncertainties obtained from the repeat spectral measurements represent the relative accuracy, and the uncertainties obtained from Allan variances of repeat measurements at fixed wavelengths represent the measurement sensitivity. From these analyses we estimate the measurement sensitivity, relative accuracy, and absolute accuracy of our experimental setup for $CH_4$ and $N_2O$ detections in air are around 10-20 pptv, 2 ppbv, and 20 ppbv, respectively. The absolute accuracy could be improved to be comparable to the relative accuracy by calibrating the measurement with standard "known" sample of ppb-level concentration and controlling accurately the laser frequency during the spectral measurements (Maity *et al.* (2017), Maithani *et al.* (2018)).

It is worth mentioning that for the measurements presented in Fig. 6, the effect of temperature fluctuation is not eliminated due to the relatively high concentration values compared to the LODs as well as relatively short measurement time. Still, the temperature fluctuation caused uncertainty of $CH_4$ and $N_2O$ concentration is presented in the determined concentration values, though this uncertainty is small and can be neglected in our case, can be corrected if necessary.

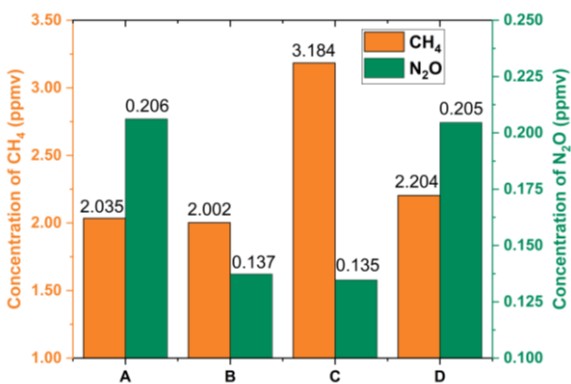

**Figure 7 Measured CH₄ and N₂O concentrations in ambient air collected at different locations (A, B, C) and in exhaled breath (D). A: Laboratory room, B: Parking lot, C: Wetland, D: Exhaled breath of one healthy person collected at the laboratory room.**

The CRDS experimental setup is then used to measure the $CH_4$ and $N_2O$ concentrations in ambient air collected at different locations and in exhaled breath of one healthy person under 1 atmospheric pressure. The results are presented in Fig. 7 and show that, (1) the $N_2O$ concentration in the in-door air of laboratory room is higher than that of out-door open fields (parking lot and wetland) (0.206 ppmv versus 0.135-0.137 ppmv). (2) The $CH_4$ concentration of out-door air collected at wetland is higher than that collected at parking lot (3.184 ppmv versus 2.002 ppmv), while the $N_2O$ concentration is little changed (0.135 ppmv versus 0.137 ppmv). This observation might be attributed to the release of $CH_4$ from the anaerobic bacteria in water and soil of wetland (Cao et al., 1998). (3) The $CH_4$ concentration of exhaled breath is approximately 169 ppbv higher than the environmental air (2.204 ppmv versus 2.035 ppmv in the laboratory room), while the change in $N_2O$ concentration is not significant (0.205 ppmv versus 0.206 ppmv). The slight variance in $CH_4$ concentration demonstrates the physiological process of $CH_4$ in human body. The concentration of $CH_4$ is closely related to some anaerobic fermentations, such as M. smithii, in human gut (Kim et al., 2012). From these measurements it is found that the measured $N_2O$ concentration of air samples, which is between 0.206 ppmv and 0.135 ppmv, is lower than the reported normal level of open air, about 0.3ppm (Davidson, 2009). This might be due to the air samples measured in our experiment are collected in a central city area, which is far away from agricultural areas where $N_2O$ is mainly produced via agricultural practices.

### 3.3 Continuous monitoring of CH₄ and N₂O in ambient air

Finally, the experimental setup is used to measured continuously the concentrations of $CH_4$ and $N_2O$ of laboratory air for 45.5 hours from 17:00PM November 6 to 14:30PM November 8, 2018. The results are presented in Fig. 8. The time resolution is 25 minutes, determined by the tuning stability of the QCL. It is experimentally observed that when the laser is tuned from one wavenumber to the next wavenumber with a step of $0.01cm^{-1}$, a time interval of approximately 6 seconds is needed to have a stable tuning (without mode-hopping). For the results presented in Fig.8, the spectrum is measured with 170 wavenumber points in the spectral band B from $1310.10cm^{-1}$ to $1311.80cm^{-1}$ with a step of $0.01cm^{-1}$. At each wavenumber the ring-down signals are recorded 150 times in approximately 7 seconds to avoid tuning instability. The laboratory air is continuously

flowing into/out the sample cell at a flow rate of approximately 2 L/min at normal atmospheric pressure. Slow fluctuations of $CH_4$ and $N_2O$ concentrations are observed due to regular air exchange (controlled by an air conditioner) between in-door laboratory air and out-door open air. It is noticed that the measured $N_2O$ concentrations are higher than that measured on June 13, 2018 and are within the reported normal range of open air, while the measured $CH_4$ concentrations are comparable to that

5    measured on June 13.

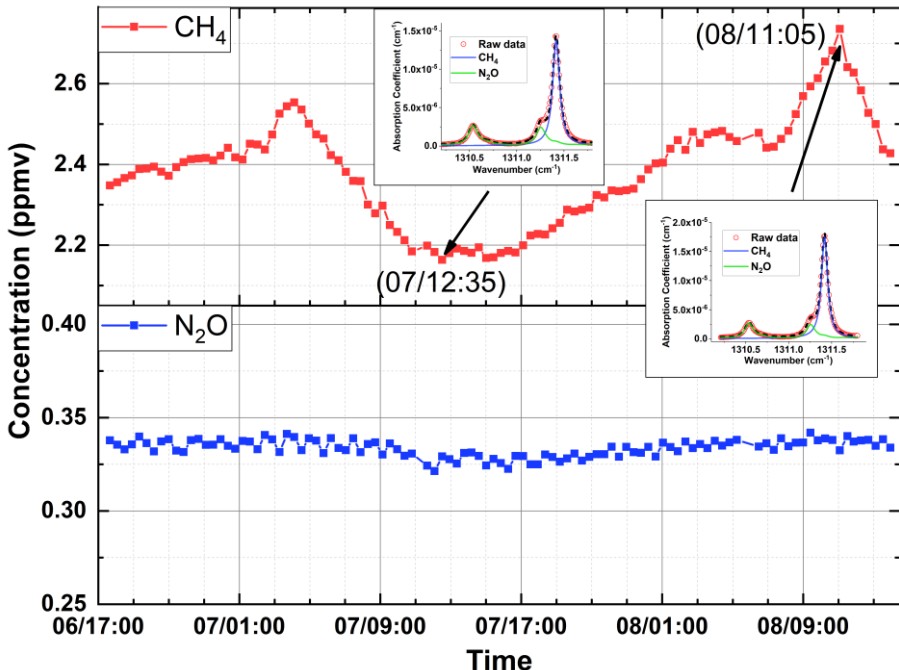

**Figure 8 Measured $CH_4$ and $N_2O$ concentrations in laboratory room (Location A) for a period of 45.5 hours from 17:00PM November 6 to 14:30PM November 8, 2018. Insets are the measured absorption spectra at two different time series and corresponding best fits for the determination of $CH_4$ and $N_2O$ concentrations.**

10    It is worth mentioning that in the measurement results presented in Figs. 7 and 8, the effect of temperature fluctuation is not eliminated, as the measurement sensitivity of CRDS experimental setup without temperature effect correction is sufficiently high that makes the correction un-necessary, as there will be no quantitative difference between uncorrected and corrected data under our experimental conditions. Still, the idea to eliminate the effect of temperature fluctuation on the trace gas detection presented in this paper is helpful to situations where very high sensitivity is required for the detection of trace gases in locations

15    where temperature is not well controlled, for example, long-term unattended out-door or wild-field monitoring of trace gases in the ppbv to sub-ppbv levels. In open fields the temperature changes widely during day and night and the effect of temperature fluctuation may become significant. The temperature effect can be eliminated by measuring the temperature dependence of measured concentrations before the CRDS instruments are placed to the wild fields. Once the CRDS instruments are in place where temperature is monitored, the temperature effect can be corrected accordingly. It is worth mentioning that the subtracting

method described in section 3.1 to eliminate the effect of temperature fluctuation is applicable only when the temperature fluctuation is periodic. In principle the effect of temperature fluctuation can always be eliminated by establishing a quantitative relation between the temperature and the ring-down time, if such a quantitative relation is experimentally repeatable and measurable. In our case, this method is not used as such quantitative relation is unfortunately very complicated.

**4 Conclusion**

We have developed an ultra-highly sensitive trace gas sensor based on Mid-IR cw-CRDS technique, in which a tunable EC-QCL at central wavelength of ~7.6 µm was employed to cover several strong absorption lines of $CH_4$ and $N_2O$. We have observed low-frequency periodical fluctuations of measured ring-down time, and correlated ring-down time fluctuations mainly to temperature fluctuations presented in test site. It was found that such correlation was attributed to creep and thermal-
drift characteristics of PZTs employed to modulate the cavity length for coupling the laser power into the ring-down cavity. By mathematically eliminating the effect of temperature fluctuation, a sensitivity limit of $7.2 \times 10^{-10}$ cm$^{-1}$ has been experimentally achieved with 21-seconds averaging time and could be further improved to $2.3 \times 10^{-10}$ cm$^{-1}$ with 519-seconds averaging time. For $CH_4$ and $N_2O$ absorption lines located at 1298.60 cm$^{-1}$ and 1327.07 cm$^{-1}$, with temperature effect correction detection limits of 13 pptv and 11 pptv were experimentally achieved with 10.4-seconds and 10.2-seconds averaging time, and
could be further improved to 5 pptv and 9 pptv by increasing the averaging time to 482.5 seconds and 311 seconds, respectively. The measurements of $CH_4$ and $N_2O$ concentrations with different spectral bands have demonstrated that $CH_4$ and $N_2O$ concentrations could be simultaneously determined at one atmospheric pressure with precision in the order of ppbv level. Finally, this CRDS setup could be easily adapted for the detections of other gases such as $C_2H_2$, $H_2O_2$, $H_2S$, $SO_2$ and sulfides with anticipated detection limits in the ppbv or even pptv level.


**Data availability.** Data are available from the authors upon request.

**Author contribution.** BL and JT designed the experiments and JT and JW performed the experiments and data processing. JT and BL prepared the manuscript with contributions from all co-authors.

**Competing interests**. The authors declare that they have no conflict of interest.

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
