# Peer review of "High-precision measurements of nitrous oxide and methane in air with cavity ring-down spectroscopy at 7.6µm"

_Atmospheric Measurement Techniques, 2018_

## Referee Comment (RC1) · Ghysels-Dubois (Referee) · 11 Dec 2018

**Ghysels-Dubois (Referee)**

melanie.ghysels-dubois@univ-reims.fr

Received and published: 11 December 2018

Review of the manuscript AMT-2018-385:

The manuscript untitled "High precision measurements of nitrous oxide and methane in air with cavity ring-down spectroscopy at 7.6um", AMT 2018-385, presents a Cavity Ring Down Spectrometer (CRDS) dedicated to atmospheric measurements of nitrous oxide and methane at room temperature. A particularity of this setup is that it works in the MIR (7.4-7.8 um). Few CRDS setups are currently operating at those wavelenghts. The described CRDS setup is claimed having a detection limit of about 7.10-10 cm-1, also, standard deviation on the ring down time is of about 0.03%, which is a standard for

CRDS setup. Here are some comments, first part are general comments and second part are technical questions.

General comment : Some English corrections are needed like for example : line 8-10, p 4 "the threshold trigger sends out a triggering signal to shutdown the AOM and a ring-down signal sequence is occurred and recorded by...". "is occured is not correct. Some other spots like this one have been detected.

Mistake line 14, p2 "fineness" is "finesse".

Please add reference "Long et al, Opt Lett. 2016 Apr 1; 41(7): 1612–1615, doi: [10.1364/OL.41.001612]" line 30 p 2. This work shows the development of MIR CRDS setup.

Specific comments :

Temperature fluctuations : line 26, p 3 : why not having picked Invar as material instead of stainless steel? Since the authors are not using frequency stabilisation scheme to correct for frequency drifts due to temperature changes. The cell body thermal expansion generates absolute frequency change of the cavity mode position and small changes of the FSR. This will results in variations in the ring down time. I agree that stainless steel is the standard material for commercial tubing and therefore easier to obtain. But Invar thermal expansion is 17 times smaller that stainless steel and so would be the frequency fluctuations. Using Invar and stabilizing the tube temperature using simple PID setup could de facto remove the help of processing correction, therefore eliminating additionnal sources of bias.

Line 11, p 4 : what is the pressure sensor used and then, it's accuracy? Then, what is the impact of the pressure sensor accuracy on the measurement uncertainty? We know that the pressure is one of the principal source for abundance measurement's accuracy.

Line 18, p 5: The authors say that the cavity is pumped down to 6.4 mbar. Is this the

lower pressure reached?

P7 : The authors shows that the ring down time is linearly dependent on the offset voltage applied to each of the 3 PZTs used. The time constant change can be as high as 10%... How are the mirrors attached to the cell? Are they glued?

Spectra acquirement: The authors do not describe how they jump for one cavity mode to another during spectra acquisition. Do they use wavelenght meter for targetting the next TEM00 mode? How stable is the frequency axis of the spectra?

Spectra processing: The authors specify that the differences between CH4 or N2O measured VMR are due to AOM-induced small change in deflection angle. Then, is this effect systematic? repeatable? How can we identify the true value from the 3 sections? Can the authors provide fit residuals along with spectra (fig 6)? The measurement differences can also be attributed to spectroscopic uncertainties or on the section B, due to the presence of N2O line beneath CH4 lines... Have these options been investigated by the authors? If so, can the author explain what are the conclusions of their investigation?

---

## Referee Comment (RC2) · Shuiming (Referee) · 12 Dec 2018

The manuscript presents a QCL-based CRDS instrument developed for ambient N2O and CH4 analysis. A moderate sensitivity of 7E-10cm-1 was obtained. A detection limit of 18 and 14 pptv respectively for CH4 and N2O was claimed with an averaging time of about 20 seconds.

I have several comments on the methods and results given in the manuscript.

(1) I am not convinced on the method used to suppress the temperature fluctuation. The authors did not give a direct correction of the data according to the temperature

(there is no simple quantitative relation between them), but simply cut off the lowfrequency components in the FFT signal of the data shown in Fig.2d, which leads to a more stable baseline in Fig.2c. The effect was "proved" to be useful as the Allan deviation (note that variance is the square of the deviation) shown in Fig.3. The new minimum in the Allan deviation appears at around n=3000 (t=400s), being consistent to the cut-off frequency of about 0.002Hz shown in Fig.2f. Is such a method applicable in a real measurement if the user does not know the exact value he/she is measuring? To any noisy spectrum, one can "remove" the low-frequency noise and "improve" the performance in respective Allan deviation. I cannot see the value of this method. At least the data presented in this manuscript is not enough to support the effectiveness of the method.

(2) The present manuscript is lack of quantitative analysis. A very high precision in the abundance was given, about 0.12% in the CH4 concentration shown in Fig.6. As far as the authors have shown in this manuscript, the absolute value of the gas concentration was not calibrated using standard/known samples. Many factors need to be clarified: How accurate is the pressure gauge (company stated value is over 1mBar)? Note the line strength data in the HITRAN database also has limited accuracy (typically 1%). Has the correction due to temperature change been included? Note that the values given in HITRAN is that at 296K. Corrections (ideal gas, population on the lower state, etc) are needed since the measurements were taken at different temperatures (and drifting!). The concentration derived from the fit of the spectrum is also dependent on the line profile used in the fitting. The profile (I guess Voigt) and the parameters used in the fitting should be explained explicitly for such a highly accurate measurement. Note the accuracy of the original parameters for these transitions.

(3) How the laser frequency is calibrated in this study? Is it good enough to support a quantitative analysis with 0.1% precision? Note that the typical line width is about 0.1cm-1, I would say, a frequency precision better than 0.001cm-1 (30MHz) is the minimum requirement for a measurement with 0.1% accuracy.

(4) It is interesting that the PZT voltage has a considerable impact on the ring-down time: Fig.4 shows a change of about 2%. Since different PZT voltage was applied to match the cavity length with the laser frequency, the voltage (and consequently the ring-down time) would be changing during the frequency scan. How this effect was included in the determination of the N2O/CH4 concentrations?

(5) The measurements shown in the study are all for samples with concentrations at the 1ppmv level, which does not support (or not enough) the 10 pptv sensitivity claimed in the manuscript. Perhaps a measurement using standard samples with much lower concentration (<10 ppbv) would be helpful.

СЗ

---

## Referee Comment (RC3) · Anonymous Referee #3 · 28 Dec 2018

The author's present a cavity ringdown spectrometer operating in the mid-infrared for measurement of CH4 and N2O. Applications for urban air and breath analysis were demonstrated. Detection limits presented for temperature-corrected measurements of target species are lower than values previously reported for MIR-CRDS in this wavelength region in literature or commercially available.

Specific Comments:

1) A more detailed discussion of the drying procedure and the impact of water vapor on the data is necessary. It would be beneficial to provide data before and after the drying procedure and to discuss in detail how data could be influenced by water vapor

(in the various spectral regions discussed here), if it is not successfully removed. Does the data analysis approach look for interferences and flag spectra if necessary?

2) Additionally, providing more information regarding the protocol for human breath analysis would be beneficial. Some questions that come to mind: what volume of sample is required to fill the cavity and how long must a participant exhale to achieve this sample volume?

3) Page 11, Line 13: Since you show that the temperature correction improves your overall detection limit, why wouldn't you implement the corrections for all scenarios? Although you indicate the system achieves sensitivity necessary without correcting for temperature fluctuations, it would be useful to state if there is a quantitative difference between uncorrected and corrected data under all experimental conditions.

4) When making assumptions or inferences regarding the cause related to your observations, include supporting literature. Two points stick out as needing further explanation or support: page 10, line 8 regarding the effect of rain on N2O retrievals and page 11, line 4 pertaining to ventilation system impacts on N2O.

Technical Corrections:

There are numerous grammar errors throughout the paper that need to be addressed. A few are listed here:

1) Page 2, Lines 4 and 9: Remove "On the other hand"

2) Page 2, Line 27: Remove "in" after 16 $\mu$W

3) Page 3, Line 5: The word "details" should be corrected to read "detail."

4) Page 3, Line 12: The word "agreements" should be corrected to read "agreement."

5) Page 4, Line 9: The phrase "is occurred and recorded" should be corrected to read "occurs and is recorded."

6) Page 5, Line 29: Consider rewording this sentence

In addition to grammar, consider the following structural changes:

1) Adding subheadings within the results and discussion section

2) Page 8, Line 20 to Page 9, Line 5: Consider using a table to describe the spectral regions. It would be easier for the reader to digest.

---

## Short Comment (SC1) · 10 Jan 2019

Dear Prof. Ghysels-Dubois

Thank you very much for your comments on our manuscript. We are currently preparing the responses to your comments. Could you please tell us the meaning of VMR (in your last comment "spectra processing")?
* * *

---

## Author Comment (AC1) · 25 Jan 2019

Please see the supplement file.

Please also note the supplement to this comment:
https://www.atmos-meas-tech-discuss.net/amt-2018-385/amt-2018-385-AC1-supplement.pdf

---

## Author Comment (AC2) · 25 Jan 2019

Responses to Second Reviewer's Comments

The second reviewer's comments are greatly appreciated. The manuscript will be substantially revised following the reviewer's suggestions. Below are the explanations to the changes will be made in the revised version and the responses to the reviewer's comments.

The manuscript presents a QCL-based CRDS instrument developed for ambient $N_2O$ and $CH_4$ analysis. A moderate sensitivity of 7E-10 $cm^{-1}$ was obtained. A detection limit of 18 and 14 pptv respectively for $CH_4$ and $N_2O$ was claimed with an averaging time of about 20 seconds.

I have several comments on the methods and results given in the manuscript.

(1) I am not convinced on the method used to suppress the temperature fluctuation. The authors did not give a direct correction of the data according to the temperature (there is no simple quantitative relation between them), but simply cut off the low frequency components in the FFT signal of the data shown in Fig.2d, which leads to a more stable baseline in Fig.2c. The effect was "proved" to be useful as the Allan deviation (note that variance is the square of the deviation) shown in Fig.3. The new minimum in the Allan deviation appears at around n=3000 (t=400s), being consistent to the cut-off frequency of about 0.002Hz shown in Fig.2f. Is such a method applicable in a real measurement if the user does not know the exact value he/she is measuring? To any noisy spectrum, one can "remove" the low-frequency noise and "improve" the performance in respective Allan deviation. I cannot see the value of this method. At least the data presented in this manuscript is not enough to support the effectiveness of the method.
Reply: We agree with the reviewer. The original thought was that we experimentally observed a correlation between the ring-down time fluctuation and the temperature fluctuation and we wanted to eliminate the effect of temperature fluctuation on the spectroscopic measurement. Cutting off the low frequency components in the FFT spectrum is an easy to eliminate the effect of temperature fluctuation, which is within the low frequency range. But as the reviewer points out, the effects of other low frequency components are also simultaneously eliminated. After careful thinking, we believe subtracting the FFT spectrum of the temperature fluctuation in the FFT spectrum of the ring-down time should be more appropriate if the temperature fluctuation is periodical, as is in our case. The results are presented in Fig.1 below. Both FFT spectra are normalized to the frequency components with the maximum amplitudes, which appears at the main frequency of the temperature fluctuation. This is reasonable if at the frequency of the temperature fluctuation, the contributions of other factors to the fluctuation of the ring-down time are negligible as compared to that of the temperature fluctuation. From the results presented in Fig. 2 below, the improvement obtained with the subtracting method is comparable to that obtained by simply cutting off the low frequency components of the FFT spectrum as we did in our original manuscript. In the revised manuscript, the cutting off method will be replaced by the subtracting method, and all results related to the elimination of the effect of temperature fluctuation will be re-calculated.

[Figure]

**Figure 1.** The "empty" ring-down time sequence recorded in a long time period (over 1 hour) and (d) corresponding FFT spectrum. (b) The synchronously recorded temperature in the laboratory room and (e) corresponding FFT spectrum. (c) The "empty" ring-down time sequence after the temperature effect is eliminated with the subtracting method and (f) corresponding FFT spectrum.

[Figure]

**Figure 2.** Allan variance plot of raw and adjusted data (temperature effect correction with the cutting off method and the subtracting method).

Worthy mentioning that even though this subtracting method may be not the most appropriate one to eliminate the effect of temperature fluctuation, it is practical in our case as only the temperature fluctuation at the detection site needs to be monitored.

In principle, the effect of temperature fluctuation can be eliminated by establishing a quantitative

relation between the temperature and the ring-down time. We also tried this method and the results are presented in Fig.3 below. Taking into account the difference of the temperature influence on the ring-down time during temperature rising and dropping periods, the data of temperature rising and dropping are fitted with third-order polynomials, respectively, and quantitative relations are established for the rising and dropping periods, respectively. Then, the ring-down time sequence is adjusted with the quantitative relations to the average temperature of 20.9 C. With this method, the absorption coefficient sensitivity limit $\alpha_{min}$ is improved to $7.0\times10^{-11}$ cm$^{-1}$, with the optimal averaging number changes to 1594. It is indicated that better results are obtained with this quantitative relation method than with the subtracting method. This method is applicable if a simple quantitative relation between the temperature and ring-down time exists and is measurable. In our case, such quantitative relation is so complicated that makes this method not practical.

[Figure]

**Figure 3. Temperature effect correction by establishing a relation between the measured ring-down time and the temperature. (a) The "empty" ring-down time and temperature recorded over 1 hour and (b) corresponding FFT spectrum. The temperature-corrected data and FFT spectrum are also presented. (c) The correlation between the ring-down time and the temperature. (d) Allan variances.**

We would like to mention that the effect of temperature fluctuation can be eliminated only if such effect can be measured and well described mathematically in some way. In some cases the effect of temperature fluctuation is very complicated and cannot be described mathematically, therefore cannot be eliminated. This point will be mentioned and more discussion on the methods of data processing for temperature effect correction will be presented in the revised manuscript.

(2) The present manuscript is lack of quantitative analysis. A very high precision in the abundance was given, about 0.12% in the $CH_4$ concentration shown in Fig.6. As far as the authors have shown in this manuscript, the absolute value of the gas concentration was not calibrated using standard/known samples. Many factors need to be clarified: How accurate is the pressure gauge (company stated value is over 1mBar)? Note the line strength data in the HITRAN database also has limited accuracy (typically 1%). Has the correction due to temperature change been included? Note that the values given in HITRAN is that at 296K. Corrections (ideal gas, population on the lower state, etc) are needed since the measurements were taken at different temperatures (and drifting!). The concentration derived from the fit of the spectrum is also dependent on the line

profile used in the fitting. The profile (I guess Voigt) and the parameters used in the fitting should be explained explicitly for such a highly accurate measurement. Note the accuracy of the original parameters for these transitions.

Reply: the reviewer is certainly right. From the viewpoint of testing instrument development, the absolute accuracy should be given by calibrating the experimental setup with standard "known" samples of comparable concentrations. Unfortunately, due to the difficulty to prepare "standard" gas mixture with known concentrations in ppb level locally, the calibration is not performed (A standard "known" sample with the lowest concentration of 1 ppmv and relative uncertainty of 2% (± 20ppbv) could be prepared by Messer Group GmbH, which is not good for our calibration purpose). Even though CRDS is in principle an absolute measurement technique therefore the CRDS-reported concentration values should represent the true values, many factors as mentioned by the reviewer affect the absolute accuracy of the measurement results. In our manuscript, the reported uncertainties of the concentration measurements are the standard deviations of multiple measurements, which represent the repeatability or the relative accuracy, not the absolute accuracy of the measurements. On the other hand, the absolute accuracy can be much improved by a calibration procedure. As the focus of our manuscript is mainly the high measurement sensitivity, not the absolute measurement accuracy, we hope it is acceptable.

To respond, in the revised manuscript, we will also present the concentration uncertainties determined from the spectral fit residuals between the measured and HITRAN spectra, which we believe represent the absolute measurement accuracy. We will discuss the sensitivity limit, relative accuracy, and absolute accuracy in details to address the accuracy issue from the Allan variances of repeat measurements at fixed wavelengths, uncertainties determined by repeat spectral measurements, and uncertainties determined from fit residuals between measured and HITRAN spectra.

(3) How the laser frequency is calibrated in this study? Is it good enough to support a quantitative analysis with 0.1% precision? Note that the typical line width is about 0.1cm$^{-1}$, I would say, a frequency precision better than 0.001cm$^{-1}$ (30MHz) is the minimum requirement for a measurement with 0.1% accuracy.

Reply: The absolute frequency of the laser is not calibrated, but linearly shifted to match the spectral lines of the target gas when performing the spectral fitting. The QCL is operated at a hop-free mode with a nominal line width <10MHz (in 1s). The frequency is tuned by the controller of the QCL (via synchronously controlling the tuning grating and the cavity length) with a step of 0.01cm$^{-1}$. At each step, the length of the ring-down cavity (RDC) is modulated via the PZTs by approximately one FSR (300MHz) to make the laser spectral line in resonance with one RDC mode. However, the frequency at each RDC mode is not accurately controlled. In our case the maximum frequency error should be <0.01 cm$^{-1}$, determined by the QCL. Due to the influence of frequency error and spectral intensity error, the precision will be around 1%. The reported 0.1% precision is the result of repeat measurements, represents a relative precision. This point will be mentioned and the influence of frequency error on the measurement accuracy will be discussed in details in the revised manuscript.

(4) It is interesting that the PZT voltage has a considerable impact on the ring-down time: Fig.4 shows a change of about 2%. Since different PZT voltage was applied to match the cavity length with the laser frequency, the voltage (and consequently the ring-down time) would be changing during the frequency scan. How this effect was included in the determination of the N$_2$O/CH$_4$

concentrations?

Reply: The linear relationship between the offset voltage on each PZT and the measured ring-down time is measured when an offset voltage is applied to only one PZT while no offset voltage is applied to the other two PZTs. When CRDS measurements are performed, the same voltage is synchronously applied to all three PZTs. In this case, the measured ring-down time is independent on the drive voltage, as Fig.4 below shows. This point will be explained in the revised manuscript.

[Figure]

Figure 4. Linear relationship between the offset voltage on each PZTs and on all PZTs simultaneously and the measured ring-down time.

(5) The measurements shown in the study are all for samples with concentrations at the 1ppmv level, which does not support (or not enough) the 10 pptv sensitivity claimed in the manuscript. Perhaps a measurement using standard samples with much lower concentration (<10 ppbv) would be helpful.

Reply: the reviewer is certainly right. The difficulty for us is to obtain "standard" gas mixture with known concentrations in ppb level locally. The best we can get is a standard "known" sample with a concentration of 1 ppmv and relative uncertainty of 2% (± 20ppbv) prepared by Messer Group GmbH. On the other hand, 10 pptv is the measurement sensitivity, which is mainly determined by the instrument's response to small change in absorption of the gas under test and is usually obtained by the standard deviation of multiple measurements. In the revised manuscript, we will discuss the sensitivity limit, relative accuracy, and absolute accuracy in details to address the accuracy issue. Our experimental setup shows a low sensitivity limit and a high relative accuracy due to the high stability of the setup, but a relatively low absolute accuracy due to the wavelength uncertainty, spectral intensity error, pressure error, etc. and lack of calibration with standard samples. The absolute accuracy can be much improved by calibration, which will normally be performed for real applications. Please also refer to our reply to comment (2).

---

## Author Comment (AC3) · 25 Jan 2019

**Responses to Third Reviewer's Comments**

The third reviewer's comments are greatly appreciated. The manuscript will be substantially revised following the reviewer's suggestions. Below are the explanations to the changes will be made in the revised version and the responses to the reviewer's comments.

The author's present a cavity ringdown spectrometer operating in the mid-infrared for measurement of $CH_4$ and $N_2O$. Applications for urban air and breath analysis were demonstrated. Detection limits presented for temperature-corrected measurements of target species are lower than values previously reported for MIR-CRDS in this wavelength region in literature or commercially available.

Specific Comments:

1) A more detailed discussion of the drying procedure and the impact of water vapor on the data is necessary. It would be beneficial to provide data before and after the drying procedure and to discuss in detail how data could be influenced by water vapor (in the various spectral regions discussed here), if it is not successfully removed. Does the data analysis approach look for interferences and flag spectra if necessary?

Reply: Due to the strong absorption of water vapor in the MIR spectral range, the water vapor has to be largely removed from the target gas. Otherwise no CRDS signals can be observed in the whole laser spectral range. Figure 1 below shows the calculated HITRAN spectra of $CH_4$, $N_2O$ and $H_2O$ at 1 atm pressure and 296K temperature. The assumed concentrations are $CH_4$: 2ppmv, $N_2O$: 0.3ppmv, and $H_2O$: 1.39% (no drying at 296K, 50% RH) and 10 ppmv (after drying. In our experiment <1ppmv is reached).

[Figure]

Figure 1. HITRAN spectra of 0.3ppmv $N_2O$, 2.0ppmv $CH_4$ and 1.39% and 10ppmv $H_2O$ in the spectral range from 1290 $cm^{-1}$ to 1350 $cm^{-1}$ at 1 atm pressure and 296K temperature.

When the water vapor is largely removed from the sample cell, for example when its concentration becomes below 10 ppmv, its influence on the measurements of $CH_4$ and $N_2O$ becomes negligible,

as Fig.2 below shows. In our case, the residual water vapor after drying is below 1 ppmv. The drying method we used can keep the water vapor below 1 ppmv in the sample cell for several months. On the other hand, the spectral line of 1312.5 cm$^{-1}$ of the water vapor, as presented in section B, can be used to monitor the water vapor concentration if necessary.

[Figure]

Figure 2. HITRAN spectra of N$_2$O, CH$_4$ and H$_2$O in the four spectral sections A, B, C, and D at 1 atm pressure and 296K temperature.

To respond, a more detailed discussion of the drying procedure and the impact of water vapor on the data will be given in the revised manuscript.

2) Additionally, providing more information regarding the protocol for human breath analysis would be beneficial. Some questions that come to mind: what volume of sample is required to fill the cavity and how long must a participant exhale to achieve this sample volume?
Reply: In our experiment, the exhaled breath air is first collected with a 3L sampling bag, which can be fully filled with only one deep breath of a participant. The filled sampling bag is then connected to the sample cell via a valve. The sample cell is first vacuumed by the vacuum pump and then filled with the exhaled air by opening the valve. This procedure is repeated two times for a complete replacement of gas in the sample cell by the exhaled air. As the volume of the sample cell is around 0.5L, the exhaled air of the 3L sampling bag is sufficient for exhaled air measurement. These details will be described in the revised manuscript.

3) Page 11, Line 13: Since you show that the temperature correction improves your overall detection limit, why wouldn't you implement the corrections for all scenarios? Although you indicate the system achieves sensitivity necessary without correcting for temperature fluctuations, it would be useful to state if there is a quantitative difference between uncorrected and corrected data under all experimental conditions.
Reply: In general, we agree with the reviewer. The purpose of presenting the temperature correction is to show in some cases the effect of temperature fluctuation on CRDS measurement

is significant and needs to be taken into consideration. We do not implement temperature correction when performing the spectral measurements to determine the concentrations of $CH_4$ and $N_2O$ in air for the following reasons: (1) In our experiment the temperature fluctuation induced uncertainty of absorption measurement is below $10^{-8}$ cm$^{-1}$, which is well below the fitting residuals between measured and HITRAN spectra, makes the correction un-necessary, as there will be no quantitative difference between uncorrected and corrected data under our experimental conditions. (2) When spectrum measurement is performed, the effect of temperature fluctuation on the measurement becomes more complicated that makes the correction more difficult. For these reasons, we prefer not to implement the temperature corrections for the spectral concentration measurements. Hope this is acceptable.

4) When making assumptions or inferences regarding the cause related to your observations, include supporting literature. Two points stick out as needing further explanation or support: page 10, line 8 regarding the effect of rain on $N_2O$ retrievals and page 11, line 4 pertaining to ventilation system impacts on $N_2O$.

Reply: The reviewer is absolutely right. Those assumptions without supporting literature will be deleted in the revised manuscript.

Technical Corrections:

There are numerous grammar errors throughout the paper that need to be addressed. A few are listed here:

1) Page 2, Lines 4 and 9: Remove "On the other hand"
2) Page 2, Line 27: Remove "in" after 16 _W
3) Page 3, Line 5: The word "details" should be corrected to read "detail."
4) Page 3, Line 12: The word "agreements" should be corrected to read "agreement."
5) Page 4, Line 9: The phrase "is occurred and recorded" should be corrected to read "occurs and is recorded."
6) Page 5, Line 29: Consider rewording this sentence

Reply: These grammar errors will be corrected in the revised manuscript.

In addition to grammar, consider the following structural changes:

1) Adding subheadings within the results and discussion section
2) Page 8, Line 20 to Page 9, Line 5: Consider using a table to describe the spectral regions. It would be easier for the reader to digest.

Reply: these two suggestions will be followed in the revised manuscript.

---

## Author Response (AR1)

**Responses to Reviewer's Comments**

The reviewer's comments are greatly appreciated. The manuscript is substantially revised following the reviewer's suggestions. The changes are highlighted in red color in the revised manuscript. Below are the explanations to the changes made in the revised version and the responses to the reviewer's comments.

**First Reviewer**

The manuscript untitled "High precision measurements of nitrous oxide and methane in air with cavity ring-down spectroscopy at 7.6um", AMT 2018-385, presents a Cavity Ring Down Spectrometer (CRDS) dedicated to atmospheric measurements of nitrous oxide and methane at room temperature. A particularity of this setup is that it works in the MIR (7.4-7.8 um). Few CRDS setups are currently operating at those wavelengths. The described CRDS setup is claimed having a detection limit of about 7.10-10 cm-1, also, standard deviation on the ring down time is of about 0.03%, which is a standard for CRDS setup. Here are some comments, first part are general comments and second part are technical questions.

General comment: Some English corrections are needed like for example: line 8-10, p 4 "the threshold trigger sends out a triggering signal to shutdown the AOM and a ring-down signal sequence is occurred and recorded by...". "is occured is not correct. Reply: The manuscript is carefully checked for English mistakes.

Some other spots like this one have been detected. Mistake line 14, p2 "fineness" is "finesse". Reply: English words are carefully checked and corrected when necessary.

Please add reference "Long et al, Opt Lett. 2016 Apr 1; 41(7): 1612–1615, doi: [10.1364/OL.41.001612]" line 30 p 2. This work shows the development of MIR CRDS setup. Reply: This paper is cited in the revised manuscript.

Specific comments:

Temperature fluctuations: line 26, p 3: why not having picked Invar as material instead of stainless steel? Since the authors are not using frequency stabilisation scheme to correct for frequency drifts due to temperature changes. The cell body thermal expansion generates absolute frequency change of the cavity mode position and small changes of the FSR. This will result in variations in the ring down time. I agree that stainless steel is the standard material for commercial tubing and therefore easier to obtain. But Invar thermal expansion is 17 times smaller than stainless steel and so would be the frequency fluctuations. Using Invar and stabilizing the tube temperature using simple PID setup could de fac to remove the help of processing correction, therefore eliminating additional sources of bias.

**Reply**: The reviewer's suggestion is certainly a good idea. But, in our case, as described in the second paragraph of section 3 and Fig.4, the effect of temperature fluctuation on the CRDS measurements is mainly caused by the different temperature dependences of the responses of the

three PZTs for cavity length modulation as well as the sensitivity of the ring-down time to the misalignment of the cavity mirrors. The cavity length fluctuation caused by temperature fluctuation via thermal expansion mismatch is negligible as compared to cavity alignment fluctuation. We have put a lot of effort to identify the cause of the temperature effect and are pretty sure about the cause. Therefore in our case the use of Invar is not necessary.

Line 11, p 4: what is the pressure sensor used and then, it's accuracy? Then, what is the impact of the pressure sensor accuracy on the measurement uncertainty? We know that the pressure is one of the principal sources for abundance measurement's accuracy.

**Reply**: The pressure sensor used in our experiment is a Keller LEX-1. The nominal absolute pressure accuracy is 0.5 mbar. Figure 1 below shows the 1311.4 cm-1 spectral intensity change (assume a Voigt spectral profile) of 2ppm CH4 when the pressure changes from 1000mbar to 999.5mbar (a change of 0.5mbar) calculated from the HITRAN database. The calculation result shows that the spectral measurement error caused by a 0.5mbar pressure error is not significant.

Figure 1. The 1311.4 cm-1 spectral line of 2ppm  $CH_4$  at pressure 1000mbar and 999.5mbar, as well as the difference calculated from the HITRAN database.

To respond, the used pressure sensor and its nominal accuracy are mentioned in the revised manuscript.

Line 18, p 5: The authors say that the cavity is pumped down to 6.4 mbar. Is this the lower pressure reached?

Reply: Yes, 6.4mbar is the lowest pressure reached in our experiment, which is determined by the capacity of the vacuum pump used in the experiment. This point is mentioned in the revised manuscript.

P7: The authors show that the ring down time is linearly dependent on the offset voltage applied to each of the 3 PZTs used. The time constant change can be as high as 10%... How are the mirrors attached to the cell? Are they glued?

Reply: The mirrors are mounted to the sample cell by screws. The linear relationship between the offset voltage on each PZT and the measured ring-down time is measured when an offset voltage

is applied to only one PZT while no offset voltage is applied to the other two PZTs. When CRDS measurements are performed, the same voltage is synchronously applied to all three PZTs. In this case, the measured ring-down time is nearly independent on the drive voltage, as Fig.2 below shows. This point is explained in the revised manuscript.

Figure 2. Linear relationship between the offset voltage on each PZTs and on all PZTs synchronously and the measured ring-down time.

Spectra acquirement: The authors do not describe how they jump for one cavity mode to another during spectra acquisition. Do they use wavelength meter for targetting the next  $TEM_{00}$  mode? How stable is the frequency axis of the spectra?

**Reply**: During CRDS measurements the QCL is operated at a hop-free mode, the nominal line width is <10MHz (in 1s). The wavelength is tuned by a controller of the QCL (via synchronously controlling the tuning grating and the cavity length) with a step of 0.01cm-1. At each step, the length of the ring-down cavity (RDC) is modulated via the PZTs by approximately one FSR (300MHz) so the laser spectral line is in resonance with one RDC mode. However, the wavelength at each RDC mode is not accurately controlled. In our case the maximum wavelength error is less than 0.01 cm-1, determined by the QCL, though the wavelength is stable as the QCL is operated at the hop-free mode, as demonstrated by the low sensitivity limits of CH4 and N2O measurements, which are determined by recording the CRDS signals at their spectral line peaks. This point is mentioned in the revised manuscript.

Spectra processing: The authors specify that the differences between  $CH_4$  or  $N_2O$  measured VMR are due to AOM-induced small change in deflection angle. Then, is this effect systematic? repeatable? How can we identify the true value from the 3 sections? Can the authors provide fit residuals along with spectra (Fig 6)? The measurement differences can also be attributed to spectroscopic uncertainties or on the section B, due to the presence of  $N_2O$  line beneath  $CH_4$  lines... Have these options been investigated by the authors? If so, can the author explain what are the conclusions of their investigation?

Reply: The differences between CH4 or N2O concentrations measured at different spectral sections are mainly due to AOM-induced change in deflection angle. This effect is systematic and

repeatable. As the RDC is aligned at 1310 cm-1, in principle the concentrations obtained from section B is close to the true values. The reviewer is certainly right that the wavelength uncertainties, spectroscopic uncertainties as well as line mixing are also sources for these differences. It is well known that collisional line mixing can significantly affect absorption spectral shapes, and line mixing cannot be neglected for accurate retrievals of atmospheric CH4 abundance (Gordon et al., 2017). The fit residuals are added to the spectral fitting and are presented in Fig.3 below.

---

## Author Response (AR2)

**Responses to comments on the revised manuscript**

The authors thank Prof. Hu very much for his comments on the original manuscript and now on the revised version. Following his comments and suggestions, the quality of our manuscript is greatly improved. The revised manuscript is further revised following his suggestions. Again the changes are highlighted in red color in the re-revised manuscript. Below are the explanations to the changes made in the version and the responses to Prof. Hu's comments.

The revised MS can be accepted for publish after a few minor corrections:
(1) A scan step of 0.01cm-1 (line 2, page 4) was given, which determines the frequency precision of the spectrum recorded by the instrument. What is the accuracy of the step control? How stable it is? It needs to be clarified.

Reply: From communications with the vendor of the QCL, the accuracy of the scan step should be better than 100MHz or 0.003 cm-1 when operated at hop-free mode. This is pointed out in the revision. In addition, the value of the laser linewidth (10 MHz or 0.00033 cm-1 in one second) is replaced by the overall linewidth (30MHz or 0.001 cm-1 long-term). In our case, during laser tuning at each step the frequency is determined by one of the RDC modes separated by free-spectral-range (FSR, 300MHz or 0.01 cm-1) of the RDC. As the scan step is also 0.01 cm-1, and the frequency at each RDC mode is not accurately controlled, the maximum frequency error should be <0.01 cm-1. This point is also mentioned in the revision. On the other hand, as the QCL is operated in mode hop-free, during operation the frequency is very stable, which can be seen from its linewidth values (<5MHz in 0.1 second, <10MHz in 1 second, and <30MHz long-term) (Unfortunately no technical data can be obtained from the vendor).

(2) As the authors described, the values of the concentrations given in the MS is actually a relative value of the relative abundance of N2O/CH4 in air. No calibration of the results has been performed. Taking into account the drift of the scan step, I doubt that it might be impossible to calibrate the value to a 0.1% precision.

Reply: Even though in principle CRDS measures the absolute absorption, the concentrations given in the manuscript should represent the absolute abundance values. However, various errors, especially the HITRAN spectral data error and laser frequency error in the concentration determination procedure make the measurement results somewhat relatively and calibration needed. The calibration corrects errors in HITRAN data but not the frequency error. Therefore, to calibrate the concentration value to a 0.1% precision, the frequency (wavelength) should also be controlled accurately. This is pointed out in the revision with publications cited.

(3) It is meaningless to assign 3 digits for all the values of the "sensitivity".

Reply: All values of the "sensitivity" are changed to 2 digits or even 1 digit when the sensitivity is in the ppt level in the revision.

[revised manuscript text omitted]

---

## Author Response (AR3)

**Responses to comments on the re-revised manuscript**

The authors thank Dr. Chen, the associate editor very much for his comments on the on the rerevised version. The manuscript is further revised following his suggestions. Again the changes are highlighted in red color in the re-revised manuscript. Below are the explanations to the changes made in the version and the responses to Dr. Chen's comments.

Comments: Regarding Figure 8, what were the time resolution used for  $CH_4$  and  $N_2O$  monitoring? what was the reason to use such long integration time? What is the comment of the authors on the "oscillation-like" variation in the time-series measurement of  $N_2O$  concentration?

Reply: The time resolution is 25 minutes, determined mainly by the tuning stability of the QCL.

It is experimentally observed that when the laser is tuned from one wavenumber to the next wavenumber with a step of  $0.01 \text{cm}^{-1}$ , a time interval of approximately 6 seconds is needed to have a stable tuning. For the results presented in Fig.8, the spectrum is measured with 170 wavenumber points from  $1310.10 \text{cm}^{-1}$  to  $1311.80 \text{cm}^{-1}$  with a step of  $0.01 \text{cm}^{-1}$ . At each wavenumber the ringdown signals are recorded 150 times in approximately 7 seconds to avoid tuning instability. Thus one spectral measurement is finished in 20 minutes. The next spectral measurement is started after 5 minutes, resulting in a time resolution of 25 minutes for simultaneous CH4 and N2O detections. The time resolution could be improved to several minutes either if the QCL has no such tuning instability problem or by optimizing the measured spectral range and scan step, or even shortened to within one minute if spectral measurements are performed only at several wavenumbers around the peaks of the CH4 and N2O absorption lines with compromise of accuracy. The time resolution is not optimized in our manuscript as it is not a major focus.

After careful thinking and further data re-processing, we finally realize that the "oscillation-like" variation of measured N2O concentration may be not real, as after data smoothing the standard deviation of the "oscillation-like" variation is around 2.5ppbv (see the figure below), comparable to the measurement uncertainty 2ppbv. In addition, no source for such "oscillation-like" variation is more likely due to measurement uncertainty. We are very sorry for this mistake and thank the associate editor very much for raising this issue and giving us the opportunity to correct it.

Accordingly, the manuscript is further revised. The time resolution is mentioned and explained. The mistake about the "oscillation-like" variation of measured  $N_2O$  concentration is corrected by deleting the description on the "fast oscillation". Figure 8 is also re-scaled to reduce the "oscillation-like" look for  $N_2O$  measurement.

[revised manuscript text omitted]